# Atomic-resolution structures from polycrystalline covalent organic frameworks with enhanced cryo-cRED

Jian Li [1,2,4], Cong Lin [1,3], Tianqiong Ma [1] & Junliang Sun [1✉]

The pursuit of atomic precision structure of porous covalent organic frameworks (COFs) is the key to understanding the relationship between structures and properties, and further developing new materials with superior performance. Yet, a challenge of how to determine their atomic structures has always existed since the first COFs reported seventeen years ago. Here, we present a universal method for ab initio structure determination of polycrystalline three-dimensional (3D) COFs at atomic level using enhanced cryo-continuous rotation electron diffraction (cryo-cRED), which combines hierarchical cluster analysis with cryo-EM technique. The high-quality datasets possess not only up to 0.79-angstrom resolution but more than 90% completeness, leading to unambiguous solution and precise refinement with anisotropic temperature factors. With such a powerful method, the dynamic structures with flexible linkers, degree of interpenetration, position of functional groups, and arrangement of ordered guest molecules are successfully revealed with atomic precision in five 3D COFs, which are almost impossible to be obtained without atomic resolution structure solution. This study demonstrates a practicable strategy for determining the structures of polycrystalline COFs and other beam-sensitive materials and to help in the future discovery of novel materials on the other.

[1] College of Chemistry and Molecular Engineering, Beijing National Laboratory for Molecular Sciences, Peking University, 100871 Beijing, China. [2] Berzelii Center EXSELENT on Porous Materials, Department of Materials and Environmental Chemistry, Stockholm University, 10691 Stockholm, Sweden. [3] Department of Mechanical Engineering, The Hong Kong Polytechnic University, 999077 Hong Kong, China. [4] Present address: Department of Fibre and Polymer Technology, School of Engineering Sciences in Chemistry, Biotechnology and Health, KTH Royal Institute of Technology, Tekninkringen 56-58, Stockholm SE-100 44, Sweden. ✉email: junliang.sun@pku.edu.cn

Since the first covalent organic framework (COF) was discovered as early as 2005 by Prof. Yaghi and co-workers[1,2], COFs have become highly sought-after materials, which have shown promising applications in molecule storage[3] and separation[4–8], heterogeneous catalysis[9–11], sensing[12–14], energy storage[15–17], optoelectronics[18], biomedical science[19,20], etc. Such diverse applications strongly rely on the various structures and the kinds of functional groups in COFs[21]. Over the past decade, a large number of COFs have been obtained but most of their crystal structures are determined by analysis of powder X-ray diffraction (PXRD) aided by Fourier transform infrared spectroscopy (FTIR), solid-state nuclear magnetic resonance (NMR), and knowledge of reticular chemistry[22–26]. Due to the very low resolution of the PXRD data for the COFs materials, different research groups even interpreted totally different topologies with the same building blocks and PXRD[27,28]. Besides, without their precise structures at atomic level, many other uncertainties still remain: (i) inaccurate atomic positions and geometric parameters; (ii) uncertainties of the degree of interpenetration and disorder in the frameworks; (iii) unknown guest arrangement; (iv) configuration uncertainties of the functional groups. Very few reported studies of COFs by single-crystal X-ray diffraction (SCXRD) have proved that the precise atomic structure will help us to understand the relationship between material structures and properties[29,30]. Due to the poor reversibility of strong covalent bonds, COFs materials are always polycrystalline, and revealing their structures has been the key bottleneck for the COFs chemistry.

Three-dimensional electron diffraction (3D ED) is an innovative technique developed as a complement to SCXRD for the structural elucidation of nanocrystals[31–33]. COF-320 is the first example whose structure was solved and refined by using rotation electron diffraction (RED, ~30 min for data collection) data from nano single crystals in 2013[34]. Because of the beam damage, even cooling the crystals to 89 K, the resolution of RED datasets of COF-320 was limited to 1.5 Å, preventing the ab initio structure determination at the atomic level. Most of COF crystals can only survive for one to two minutes under the electron beam, and the total angular coverage is ~60° with ~33.3% of the reciprocal space sampling volume using the advanced continuous mode of data collection, which results in limited completeness and redundancy of ~40% and ~0.5, respectively[35]. Although the 3D ED technique that we used has been reported for the structure solution of 3D COFs[18,36–42], it is far from routine and still a great challenge to obtain the atomic resolution of framework structure in COFs, let alone dynamic structures with flexible linkers, determination of the degree of interpenetration, position of functional groups, arrangement of guest molecules, and disorder in frameworks with atomic precision.

Herein, we demonstrate the enhanced cryo-continuous rotation electron diffraction (cryo-cRED) method to address the beam damage issues of 3D COFs (Fig. 1). By combining hierarchical cluster analysis (HCA)[43,44] with cryo-EM technique, the 3D ED data quality regarding the resolution and completeness is improved significantly, which results in 0.79-angstrom resolution with more than 90% completeness. Using this method, structures of five complex 3D COFs (E-FCOF-5, C-FCOF-5, 3D-TPB-COF-OMe, 3D-TPB-COF-Me, and 3D-TPB-COF-OH), including a series of 3D COFs with different functional groups, different degrees of interpenetration, and dynamic structures, are determined successfully at the atomic level by the ab initio method. All the non-hydrogen atoms in the framework, together with the functional groups and guest molecules, are located directly from the electrostatic potential map. Our achievement in this study represents a universal and superior method of ab initio structure determination of COFs at atomic level. It will definitely help us

for better understanding the relationship between structure and property of COFs, and further developing new COFs with superior performance.

## Results

**3D flexible COFs with dynamic structures.** The applicability of enhanced cryo-cRED was first demonstrated on a flexible 3D COF (FCOF-5), which is formed by the [4 + 4] imine condensation reaction between the molecules of 1,2,4,5-tetrakis[(4-formylphenoxy)methyl] benzene (TFMB) with flexible C–O single bonds in the backbone and the rigid tetra(p-aminophenyl) methane (TAPM) (Supplementary Figs. 1a, 2a, and 3a). Due to the bond flexibility, this COF can undergo reversible structural expansion/contraction in response to guest molecular adsorption/desorption, indicating a breathing behavior. Although the simulated structures using PXRD were reported in our recent study[38], the dynamic structure at atomic level is still unknown. As FCOF-5 is sensitive to the environment, the cRED sample preparation should be careful. For the sample preparation of the expanded FCOF-5 (E-FCOF-5), a small quantity of micro-crystals was dispersed in ethanol by ultra-sonication for 5 min, and then a suspension droplet was transferred onto a copper grid covered with carbon film. From the grid, thousands of nanocrystals were easily discernible on the grid surface, providing ample samples for the 3D ED data collection (Supplementary Fig. 4a). Prior to being transferred into the transmission electron microscope (TEM), the sample was cooled to ~173 K using a cryo-transfer tomography holder to fix the guest molecules within the COF pores. After transferring the grid to the electron microscope, the sample was further cooled down to 96 K for 3D ED data collection. To achieve high diffraction resolution, the spot size and exposure time were optimized to 3 and 0.5 s, respectively. Although the E-FCOF-5 nanocrystals could diffract to a high resolution of ~0.82 Å, their crystallinity deteriorated quickly and the diffraction resolution dropped to ~3 Å at the final stage because of the serious beam damage (Supplementary Figs. 4 and 5). From an individual dataset (dataset 1 in Supplementary Table 1), the unit cell parameters can be easily determined to be $a = 14.58$ Å, $b = 8.51$ Å, $c = 26.49$ Å, and $\beta = 92.97°$ with monoclinic symmetry. The reflection conditions from specific slices were extracted as $h0l$: $l = 2n$, $00 l$: $l = 2n$, suggesting two possible space groups of $P2/c$ (No. 13) or $Pc$ (No.7) (Supplementary Fig. 5). Due to the completeness and redundancy were as low as 33% and 0.88, respectively, for an individual dataset, it failed to solve the E-FCOF-5 structure using ab initio methods (direct methods, charging flipping, etc.).

To improve the data completeness and redundancy for structure determination and refinement, the enhanced cryo-cRED method, which is hierarchical cluster analysis (HCA)[43,44] on several cryo-cRED datasets with different orientations of crystals, was employed. Twenty-two cRED datasets were collected on the E-FCOF-5 crystals with a large tilt range from 34.5º to 95.2º (detailed information of each dataset is listed in Supplementary Table 1). The 3D reciprocal lattices reconstructed from the twenty-two datasets are shown in Supplementary Fig. 6, where the reciprocal lattice with a high resolution is in different locations owing to the different crystal orientations. Because E-FCOF-5 belongs to the monoclinic crystal system, none of the cRED datasets can have a completeness >90%, which is necessary for a feasible structure refinement. Therefore, several datasets were carefully selected and merged. Before merging the datasets, all the cRED datasets were processed using an automated data procession method (details are stated in the "Method" section). The twenty-two cRED datasets could be indexed with an average lattice parameters of $a = 13.7095(2)$ Å, $b = 8.633(6)$ Å,

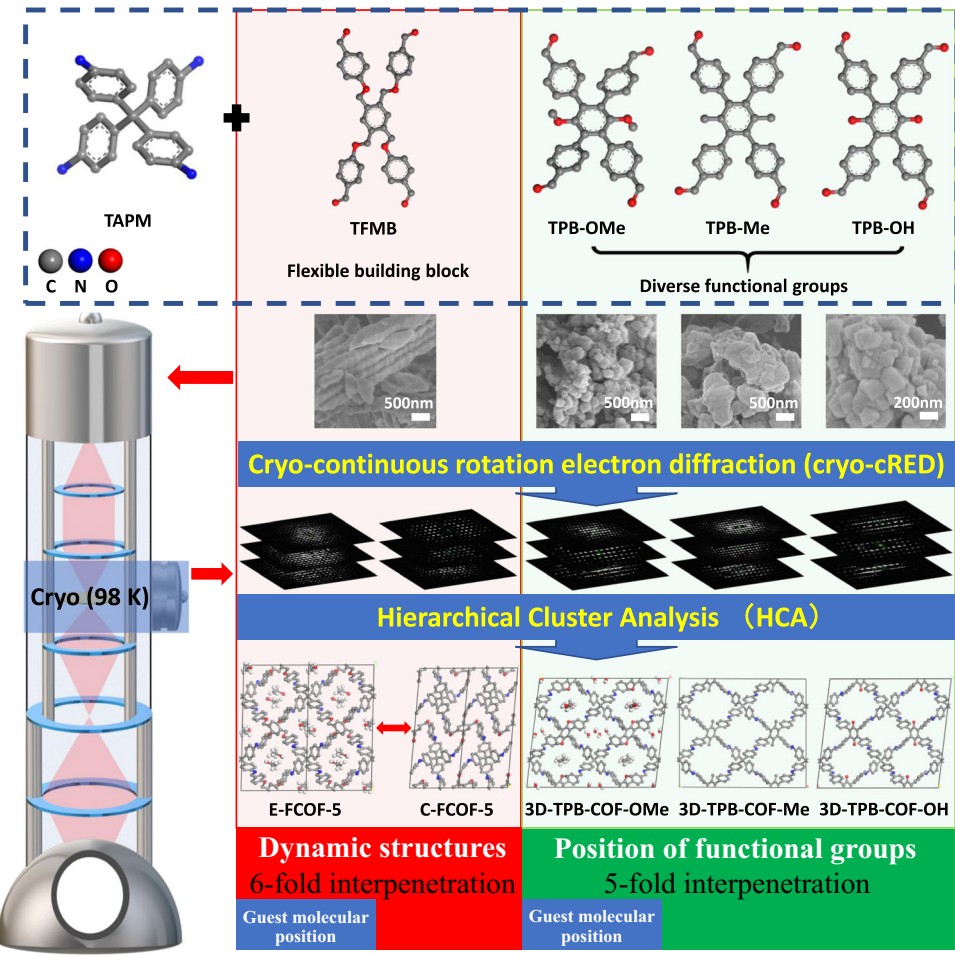

**Fig. 1 Chemical information and workflow of enhanced cryo-cRED for the structure determination of polycrystalline 3D COFs.** TAPM: tetra(*p*-aminophenyl)methane, TFMB: 1,2,4,5-tetrakis[(4-formylphenoxy)methyl] benzene, TPB-OMe, TPB-Me, and TPB-OH: methoxy-, methyl-, and hydroxy-functionalized 1,2,4,5-tetrakis(4-formylphenyl)benzene on the 3- and 6-positions.

$c = 26.4086(1)$ Å, and $\beta = 91.393(2)°$. The twenty-two datasets were then subjected to the hierarchical cluster analysis (HCA) using a Python script developed in-house to determine the optimal datasets for merging[44]. The distance metric *t*, which defines the similarity between datasets, is derived from the correlation coefficients of the common reflection intensities ($CC_I$) in dataset pairs, and the "average" linkage method is employed. The clusters can be visualized using a dendrogram, making it easier to find an appropriate cut distance. Clusters with $t < 0.40$, in our experience with HCA, usually result in usable datasets. But in the E-FCOF-5 case, the distance metric value between each dataset is a little bit larger, which appears to be due to the beam damage. We thus cut the distance metric value with $t = 0.5$ to balance the completeness and obtained three clusters (Fig. 2a). The largest one (Fig. 2a, in red), consisting of sixteen datasets (Supplementary Fig. 6, Supplementary Table 1), possessed the highest data quality with the completeness and redundancy of 91% and 9.22, respectively. The sixteen datasets belonging to the largest cluster were merged. With such a high-quality merged dataset (Fig. 2b), all non-hydrogen atoms were located directly from the electrostatic potential map by using SHELXT[45], resulting in a sixfold interpenetrated **pts** topology (Fig. 2f, left of Supplementary Fig. 7). Finally, the structure model was refined isotropically using soft restraints for the geometry of the phenyl rings as well as the C–C and C=N bond lengths. The guest molecules of ethanol can be determined by the difference electron density map (Fig. 2e, Supplementary Fig. 21a), benefited from the

high data completeness and resolution. With the atomic precision structure, the precise pore size of the expanded FCOF-5 is obtained to be 6.2 Å × 6.2 Å and 4.2 Å × 7.2 Å (Fig. 2f), which is unknown in previous study. The details of cryo-cRED experimental parameters, crystallographic data, and structure refinement are in Supplementary Table 2.

For the contracted FCOF-5 (C-FCOF-5), the crystals were placed directly onto the copper grid without any dispersion. To ensure that the framework is fully contracted, the sample was transferred into the TEM with a high vacuum ($<2 \times 10^{-5}$ Pa) at room temperature for 5 min to extract the guest molecules. After that, the sample was cooled to 96 K for data collection. Twenty-two cRED datasets (Supplementary Table 3) were obtained on the C-FCOF-5 crystals, among which sixteen were selected for merging after HCA with the distance metric *t* of 0.56 (Fig. 2c, Supplementary Fig. 8). The C-FCOF-5 unit cell in the space group $P2/c$ shrinks to $a = 10.9511(2)$ Å, $b = 7.7945(1)$ Å, $c = 26.8214(6)$ Å, and $\beta = 95.247(3)°$ (Supplementary Fig. 9) with a large volume contraction of ~27%, indicating a significant breathing motion. After similar data processing with E-FCOF-5, the completeness and redundancy for the C-FCOF-5 data were increased from 20~50% and ~1.0 (individual dataset) to 90% and 10.0 (merged sixteen datasets after HCA), respectively. The cRED data collected on the C-FCOF-5 crystals had a resolution of up to ~0.81 Å (Supplementary Fig. 10), and all the non-hydrogen atoms could be located directly from the electrostatic potential map using the ab initio structure solution with SHELXT[45], yielding the

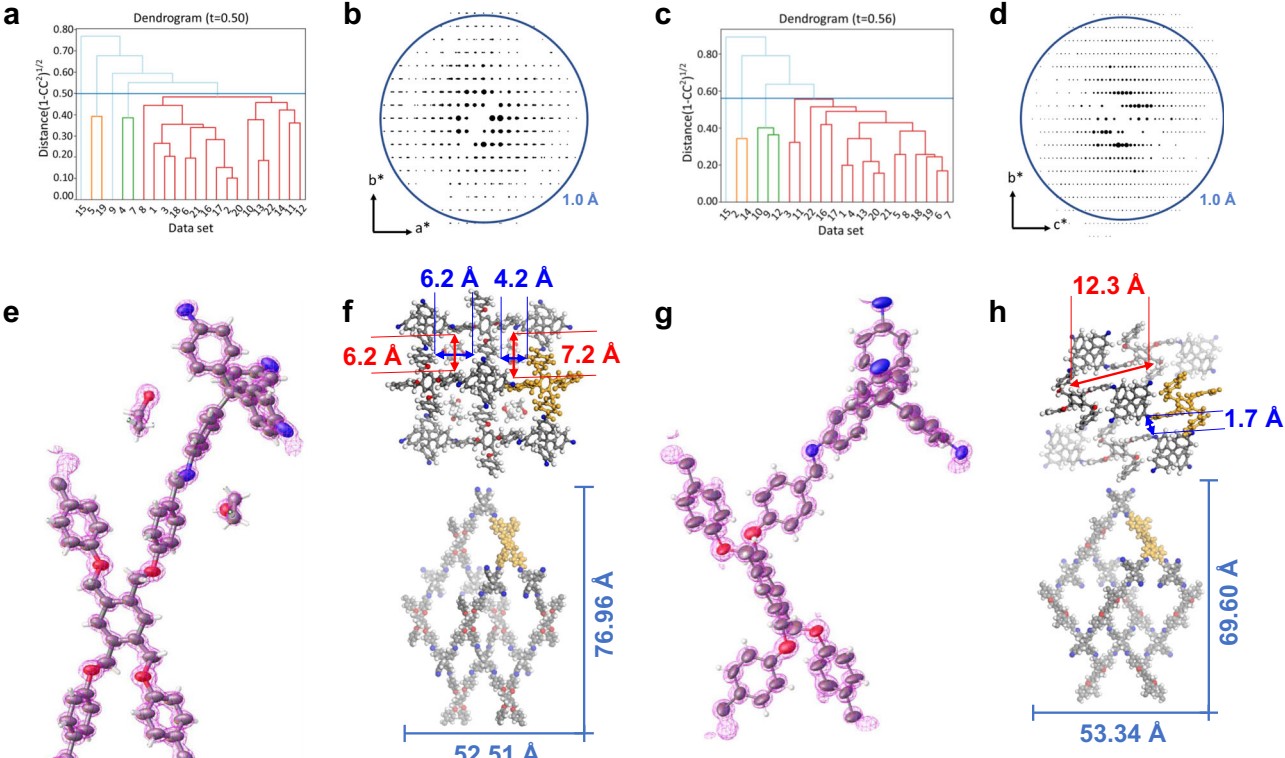

**Fig. 2 HCA and structure of E-FCOF-5 and C-FCOF-5.** The cut distance of HCA is represented in dendrogram by the blue line at 0.5 for E-FCOF-5 (**a**) and 0.56 C-FCOF-5 (**c**), which results in 16 cRED datasets belonging to the largest cluster for E-FCOF-5 (**a**, in red) and E-FCOF-5 (**c**, in red). The overview of 3D reciprocal lattices of E-FCOF-5 (**b**) and C-FCOF-5 (**d**) merged from 16 cRED datasets. The observed potential density maps of E-FCOF-5 (**e**) and C-FCOF-5 (**g**). The structures were refined isotropically using soft restraints for the geometry of the phenyl rings as well as the C–C and C=N bond lengths. The observed peaks appeared to be spherical with similar peak heights for the same atom types. The porous structure of E-FCOF-5 (**f**-up) and C-FCOF-5 (**h**-up). The single **pts** net of E-FCOF-5 (**f**-down) and C-FCOF-5 (**h**-down).

same six-fold interpenetrated *pts* topology with E-FCOF-5 (Fig. 2h, right of Supplementary Fig. 7). The C-FCOF-5 structure was also refined isotropically with soft restraints on the phenyl ring geometry as well as the C–C and C=N bond lengths and the observed peaks also appeared to be spherical with similar peak heights for the same atom types (Fig. 2g, Supplementary Fig. 21b). As expected, the building block TFMB in the contracted FCOF-5 structure is twisted with its phenyl rings blocking the pores, thus no solvent void is accessible during the PLATON/SQUEEZE procedure[46]. From the atomic precision structure, the width of the pores of contracted FCOF-5 is only 1.7 Å (Fig. 2h), which finally reveals the reason for no $N_2$ and Ar adsorption in our previous study. In addition, the bond angle and geometry of E-FCOF-5 and C-FCOF-5 obtained from simulation of PXRD data show a large deviation comparing to the atomic precision structure (Supplementary Fig. 11), indicating that atomic-level structure determination is very important to understand the dynamics of flexible COFs. To our best knowledge, it is the first time that the structures of a flexible 3D COF at both the expansion and contraction states are resolved with atomic precision. The details of cryo-cRED experimental parameters, crystallographic data, and structure refinement are in Supplementary Table 4.

**3D COFs with functional groups**. The internal functional groups of 3D COF can be used to provide a structurally accurate application platform. Encouraged by the exciting results of FCOF-5, the applicability of enhanced cryo-cRED was employed on three 3D-TPB-COFs with different functional groups of methoxy (-OMe), methyl (-Me), and hydroxyl (-OH) to thoroughly

explore the scope and applicability of this powerful ab initio structure determination method (Supplementary Fig. 1b, 2b, and 3b-d). These three 3D-TPB-COFs were synthesized by the [4 + 4] condensation reaction of three 1,2,4,5-tetraphenylbenzene (TPB) derivatives, namely TPB-OMe, TPB-Me, and TPB-OH functionalized by the respective methoxy, methyl, and hydroxyl groups, with tetra(p-aminophenyl)methane (TAPM). The synthesis condition was described in our previous reports but the single-crystal structures with the atomic resolution are unclear[36,40,41].

For 3D-TPB-COF-OMe, the nano-crystals could diffract to a high resolution of ~0.87 Å at the beginning of the cryo-cRED[32,47] data collection (Supplementary Fig. 12), but the highest completeness and redundancy from a single dataset were ~65.9% and 1.55 (dataset 4# in Supplementary Table 5, Supplementary Fig. 13), respectively, resulting in missing observable peaks, significant variations in peak heights, and severe peak elongation during refinement (Supplementary Fig. 14). Eleven cRED datasets were collected on 3D-TPB-COF-OMe crystals (Supplementary Table 5). The distance metric value is cut with $t = 0.3$, equivalent to $CC_I = 0.95$, which results in three clusters (Fig. 3b). The largest one consisting of six datasets (Fig. 3a), possessed the highest data quality with the completeness and redundancy of 91.8% and 8.43, respectively. With such a high-quality merged dataset (Fig. 3c), all non-hydrogen atoms, including the functional group of -OMe, were located directly from the electrostatic potential map by using SHELXT[45]. The structure refinement was greatly improved, from which the observed peaks appeared to be spherical with similar peak heights for the same atom types (Fig. 4a). The ordered guest molecules of water and ethanol in the pores of 3D-TPB-COF-OMe were

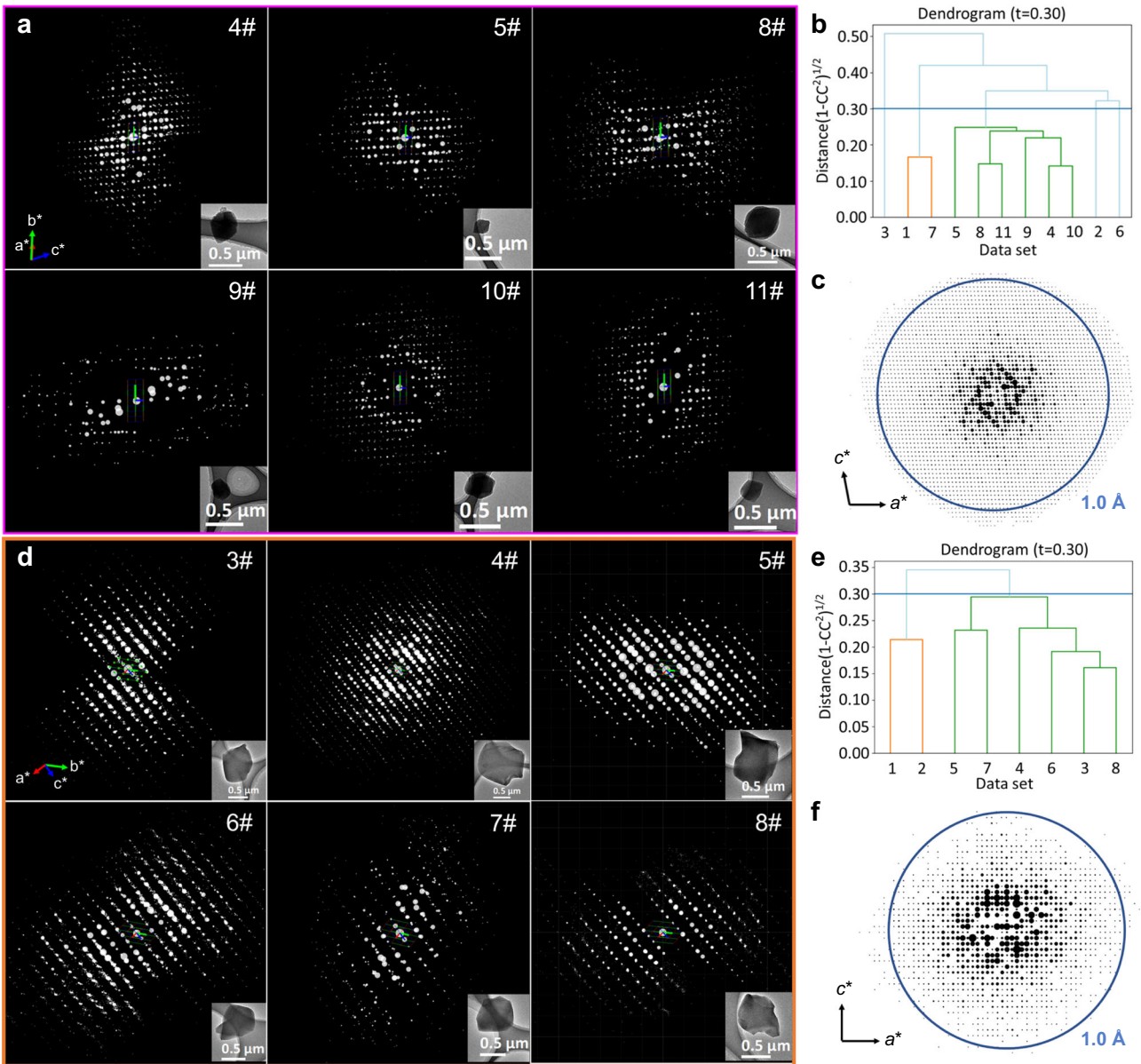

**Fig. 3 3D ED and HCA of 3D-TPB-COF-OMe and 3D-TPB-COF-Me.** The overview of automatically selected six 3D ED datasets of 3D-TPB-COF-OMe (**a**) and 3D-TPB-COF-Me (**d**) for merging and scaling. The dendrogram shows the results of the hierarchical cluster analysis (HCA) of extracted intensities using the correlation coefficients of the common reflection intensities ($CC_I$) between pairs of datasets. The cut distance is represented by the blue line at 0.30 (corresponding to $CC_I = 0.95$) for 3D-TPB-COF-OMe (**b**) and 3D-TPB-COF-Me (**e**). In total, the largest clusters were identified with six cRED datasets for 3D-TPB-COF-OMe (**b**, in green) and 3D-TPB-COF-Me (**e**, in green), which were selected for the structure determination and refinement. The overview of merged 3D reciprocal lattices of 3D-TPB-COF-OMe (**c**) and 3D-TPB-COF-Me (**f**).

observed at the final stage of refinement (Supplementary Fig. 21c). In the end, the *R1* value was converged to 0.2281. With the same method, the atomic precision structure of 3D-TPB-COF-Me (Fig. 4b) and 3D-TPB-COF-OH (Fig. 4c) were determined directly with precise anisotropic refinement by merging 6 datasets (Supplementary Table 7, Fig. 3d–f, Supplementary Figs. 15, 16, and 21d) and 4 datasets (Supplementary Table 9, Supplementary Figs. 17–19 and 21e) after HCA, respectively. 3D-TPB-COF-OMe, 3D-TPB-COF-Me, and 3D-TPB-COF-OH are determined to have a five-fold interpenetrated structure with a reported **pts** net and the similar one-dimensional (1D) straight channels, but offering varied pore sizes (Fig. 4, Supplementary Fig. 20). In addition, the position of functional groups can be refined without any restraints. The obtained bond length of C–O(–C), (C=)C–C and (C=)C–O in methoxy, methyl, and hydroxyl are 1.41, 1.55,

and 1.28 Å, respectively, which are close to the theoretical values (Fig. 4). The details of cryo-cRED experimental parameters, crystallographic data, and structure refinement are in Supplementary Tables 6, 8, and 10.

## Discussion
In summary, we present a powerful method for ab initio structure determination for sub-micron-sized crystals of 3D COFs with cryo-cRED at the atomic level. The key of this study is the application of HCA, which contributes to the optimal selection of datasets for structure solution and refinement. And after merging the best-selected datasets, the quality regarding the data completeness and redundancy can be greatly improved. As proof of such a powerful method, five complex 3D COF structures are successfully solved at the atomic level using ab initio method, in

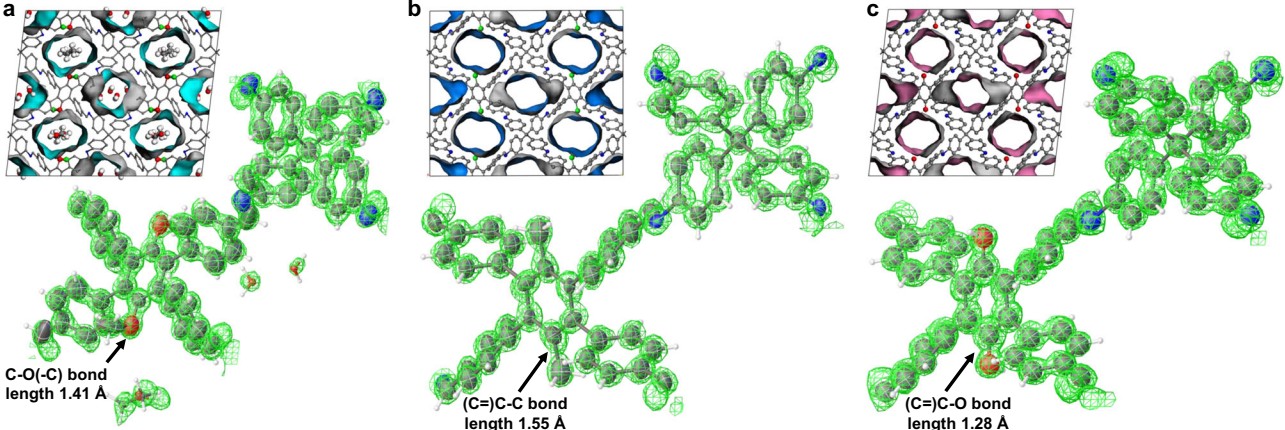

**Fig. 4 Observed potential density maps and crystal structures. a** 3D-TPB-COF-OMe, **b** 3D-TPB-COF-Me, and **c** 3D-TPB-COF-OH. Because of the high resolution and high completeness of the dataset after HCA, not only the framework atoms can be located directly, but also the functional groups and guest molecules can be determined with atomic precision by ab initio method. Color scheme: C in the framework, gray; O, red; N, blue; C in the functional groups green.

which not only all the non-hydrogen atoms in the framework are located directly from the electrostatic potential map, but the dynamic structures with flexible linkers, degree of interpenetration, position of functional groups, and arrangement of guest molecules are successfully revealed with atomic precision. Unlike other direct space methods for the 3D COF structure solution, the ab initio method does not require any knowledge of reticular chemistry. Although the emphasis of this study is on the 3D COF chemistry, the structure solution method mentioned here can also be applied to other beam-sensitive materials, such as the small organic molecules, polymers, and organic-inorganic hybrid compounds.

## Methods

**Material syntheses.** Samples of the FCOF-5, 3D-TPB-COF-OMe, 3D-TPB-COF-Me, and 3D-TPB-COF-OH were synthesized as described in the previous reports[36,38,40,41]. In short, monomers for these COFs were firstly synthesized briefly as follows. TFMB was prepared by reflux of a suspension containing 1,2,4,5-tetrakis(bromomethyl) benzene, *p*-hydroxybenzaldehyde, $K_2CO_3$, and $CH_3CN$ with specific contents for 18 h, followed by filtering and washing with water and methanol. A suspension of a certain amount of 1,2,4,5-tetrabrom-3,6-dimethoxybenzol, 5,5-dimethyl-2-[4-(4,4,5,5-tetramethyl-1,3,2-dioxaborolan-2-yl)phenyl]−1,3-dioxan, Pd(dppf)$Cl_2$·$CH_2Cl_2$, and $Cs_2CO_3$ in toluene/$H_2O$ solvent was heated in $N_2$ at 90 °C for 3 days to obtain the TPM-OMe precursor, which was then reacted in trifluoroacetic acid/$CH_2Cl_2$ solvent and followed by adding $Na_2CO_3$ soluiton to pH = 10. The mixture was then extracted, washed, and dried by $CH_2Cl_2$, brine, and anhydrous $Na_2SO_4$, respectively, to obtain the TPB-OMe. TPB-Me was synthesized from the mixture of 1,2,4,5-tetrabromo-3,6-dimethyl-benzene, 4-formylphenylboronic acid, palladium tetrakis(triphenylphosphine), $K_2CO_3$, and 1,4-dioxane/$H_2O$ with certain amounts in $N_2$ at 90 °C for 3 days. TPB-OH was prepared following a similar procedure as that of TPB-OMe with different reactants. Then, [4 + 4] imine condensation reaction between the molecules of TAPM and TFMB, TAPM and TPB-OMe, TAPM and TPB-Me, and TAPM and TPB-OH was conducted in the solvent of mesitylene, $CHCl_3$, and AcOH (in volume ratio of 7:3:1) at 110 °C for 7 days, in the solvent of $CHCl_3$, *n*-butanol, and AcOH (in volume ratio of 2:7:1) under vacuum at 120 °C for 7 days, in the solvent of *o*-dichlorobenzene, *n*-butanol, and 6 M acetic acid under vacuum at 110 °C for 7 days, and in the solvent of mesitylene, *n*-butanol, and 12 M acetic acid under vacuum at 120 °C for 7 days to synthesize the FCOF-5, 3D-TPB-COF-OMe, 3D-TPB-COF-Me, and 3D-TPB-COF-OH, respectively.

**TEM sample preparation.** For E-FCOF-5, 3D-TPB-COF-OMe, 3D-TPB-COF-Me, and 3D-TPB-COF-OH, TEM samples were prepared by firstly crushing the powder followed by dispersion in ethanol and ultrasonic treatment for 5 min. A drop of the dispersion was then deposited onto a 3 mm wide TEM copper grid covered by carbon film. A high-tilt cryogenic sample holder equipped with temperature monitor (Gatan Company, Model 914) was used to freeze the sample to ~173 K to fix the guest molecules in pores before transferring the grid into TEM. The samples were further cooled to 96 K after inserting the holder into TEM for data collection. For C-FCOF-5, the tiny crystals were loaded directly onto the copper grid without dispersion in order to avoid adsorption of any guest molecules.

To ensure that the framework is completely contracted, the sample was transferred into the TEM with high vacuum (<2 × 10$^{-5}$ Pa) at room temperature for 5 min to further remove the guest molecules, which was then cooled to 96 K for data collection. All the data collection starts when the temperature of the sample was stable at 96–99 K.

**3D ED data collection.** The continuous rotation electron diffraction (cRED) data were collected on the JEOL JEM-2100 transmission electron microscope (TEM, *Cs*: 1.0 mm, point resolution: 0.23 nm) at 200 kV using the instamatic script (https://github.com/instamatic-dev/instamatic). During the data collection, the goniometer was rotated continuously while the selected-area ED patterns were captured from the individual crystal simultaneously by a quad hybrid pixel detector QTPX-262k (512 × 512 pixels with the size of 55 μm, Amsterdam Sci. Ins.). To track the crystals during data collection, the diffraction patterns were defocused at the interval of every 10 frames, and an image was taken with a short exposure (typically 0.01 s) to check the crystal position. The rotation speed of 0.45°/s and rotation step of 0.23° were used for each dataset, and all the ED patterns were recorded with the spot size of 3 and exposure time of 0.5 s.

**Data processing.** The 3D reciprocal lattice was reconstructed using the software REDp[47] (https://sites.google.com/view/xiaodong-zous-group/software/red), which was powerful for indexing and obtaining the reflection conditions. Data processing was conducted using the software package XDS[48] (https://xds.mr.mpg.de/). As a large number of cRED datasets need to be processed, an automated data processing pipeline was used in this study, details of which can be found in our previous work[43,44]. Shortly, the data processing pipeline consists of a set of Python scripts, including functions for automatically running XDS on all the datasets, extracting the lattice parameters and integration statistics, and cluster analyses. XDS runs for all the cRED dataset folders containing the file XDS.INP. If the dataset can be indexed successfully, the lattice parameters and integration statistics are then extracted from the file CORRECT.Lp. Then, hierarchical cluster analysis (HCA) is performed to find the most common unit cell using the lattice-based clustering method and to select the optimal datasets for merging using the reflection-based clustering method. XSCALE (https://xds.mr.mpg.de/), one part of the XDS package, is used for merging the datasets and generating the .hkl files that are used for structure solution and refinement.

**Ab initio structure solution and refinement.** The SHELX[49] (http://shelx.uni-ac.gwdg.de/SHELX/) software package was used for structural analysis, where SHELXT was used for structure solution and SHELXL for structure refinement. Atomic scattering factors for electrons based on the neutral atoms were used. All the atoms were refined anisotropically. DFIX constraints were used to maintain reasonable C–C and C–N distances during the refinement. PLATON/SQUEEZES[46] (http://www.cryst.chem.uu.nl/platon/) procedure was conducted to deduct the diffraction contribution from the disordered guest molecules within the pores at the final stage of refinement for 3D-TPB-COF-Me and 3D-TPB-COF-OH.

## Data availability

All the data supporting the findings of this study are available within the Article and its Supplementary Information, or from the corresponding author J.S. (junliang.sun@pku.edu.cn) upon request. Crystallographic data for the structures

reported in this Article have been deposited at the Cambridge Crystallographic Data Center, under deposition numbers CCDC 2115021 (E-FCOF-5), 2115022 (C-FCOF-5), 2115023 (3D-TPB-COF-OMe), 2115024 (3D-TPB-COF-OH), and 2115025 (3D-TPB-COF-Me). Copies of the data can be obtained free of charge via https://www.ccdc.cam.ac.uk/structures/.

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

## Acknowledgements
J.S. gratefully acknowledges financial support from the National Natural Science Foundation of China (U21A20285, 22125102, 21871009, 21527803), and J.L. thanks the Swedish Research Council (VR, grant No.1444205) and the Knut and Alice Wallenberg Foundation (KAW, 2012-0112) through the 3DEM-NATUR project. The authors thank Dr. Xiaoling Liu, Chao Gao, and Yang Xie in Prof. Cheng Wang group (Wuhan University) for providing the 3D COF samples.

## Author contributions
J.S. and J.L. designed the project. J.L. performed cRED experiments and solved the crystal structures. J.S. supervised the experiments. J.L., C.L., T.Q.M., and J.S. prepared and revised the manuscript draft together.

## Competing interests
The authors declare no competing interests.
