## [Peer Review File · Nature Communications]

Atomic-resolution structures from polycrystalline covalent organic frameworks with enhanced Cryo-cREDReviewers' Comments:

Reviewer #1:

Remarks to the Author:

The authors developed enhanced cryo-cRED by combining hierarchical cluster analysis with cryo-EM for atomic level structural analysis of several 3D COFs. High quality 3D ED data were collected to obtain the ab initio structure solutions of five complex 3D COFs. All of non-hydrogen atoms were successfully located. This work is well presented and will benefit the development of COFs. I recommend the publication of this work after addressing the following concerns.

1. How was the cryo sample prepared? On page 5, "the sample was cooled to ~173K using a...". How was it achieved and what is the purpose? More detailed information should be provided.
2. There is a big difference among unit cell parameters from different datasets and Rint. What's the reason? How to deal with this during the merging of multiple datasets?
3. In the title, polycrystalline has been used. There is no need to use crystals afterwards. I think "Atomic-resolution structures from polycrystalline covalent organic frameworks with enhanced Cryo-cRED" is enough.
4. "ab initial" should be "ab initio".
5. In the abstract, anisotropic refinement is not a proper word. Refinement of anisotropic temperature factors or ADPs?
6. "The pursuit of...is to understand" may be changed to "The pursuit of...is a key to understand..."
7. "With such powerful method" change to "With such a powerful method".
8. Page 3, "(iv) uncertainties configuration" should be "uncertain configuration or configuration uncertainties".
9. "Due to the irreversible strong covalent bond properties," may be "due to the poor reversibility of strong covalent bonds"? Please confirm this.
10. Page 4, "gust molecular" should be "guest molecule"?
11. On page 5, "To achieve high diffraction resolution, the spot size and exposure time were optimized to 3 and 0.5s." The electron dose of spot size 3 is not low, why not using spot size 5 considering the reduce of beam damage? What is the rough electron dose or electron dose rate under this condition?
12. Unit cell parameter symbols should be italic, such as a, b, c, and beta.
13. "direct method" should be "direct methods".
14. On page 6, "The twenty-two cRED datasets could be...a = 13,7095(2)...", unit cell parameters have four decimal points and uncertainties were give. How can these parameters be so precise?
15. Could the number of ethanol molecules be precisely confirmed? Are there any unique interaction between ethanol and the framework?
16. On page 8, "To ensure that..." Could this process completely remove the guest molecules?
17. On page 11, "the atomic precession structure" should be "the atomic precision structure".
18. As the reported standard Cryo-EM sample preparation, standard fast-cooling method can keep the room-temperature structure (J. Am. Chem. Soc. 2019, 141, 19983–19987; doi:10.1038/s41467-022-29332-2). Can this cooling process keep the innate structure? As a flexible COF, dose cooling temperature influence the structure, especially for C-FCOF-5?
19. On page 14, some spelling errors, such as "need to be process", "is run", "for merge" et al. Please correct these.
20. Please check the manuscript carefully and correct the typos.

Reviewer #2:

Remarks to the Author:

This is an important contribution towards the structural determination of COF materials. This is among the most challenging structural analyses. Porous COF materials have attracted much interests since the discovery for a wide range of applications. However, due to their poorly crystalline nature, highly dynamic bonds/framework flexibility, unambiguous structural characterisation of these materials has been extremely difficult, if not possible. The growth of single crystals of COFs that are suitable for

routine single crystal diffraction has been only a couple of examples, while analysis by powder diffraction is limited by its intrinsic low resolution and heavy overlap of peaks, which is particularly the case for COFs given their complex structures. The present study has developed Cryo-cRED to determine the crystal structures of a series of complex COFs. I am impressed by the quality of the datasets and refined structures; these are almost comparable with routine single crystals with resolutions down to 0.8 Å and completeness above 90%. This is a major achievement considering the polycrystalline nature and sensitivity to beam damage of these materials!

The structural analysis has been described in great details and new structural information revealed for some of these previously known COFs. I am particularly pleased to see that the team were able to observe isolated guest molecules in the pore by this technique. This I think is a big step forward in the structural characterisation of organic polymers. I therefore strongly recommend for publication of this work in Nature Commun.

One minor point, the authors could consider to move the tables of refinement details from the SI to the main paper. This contains some important structural information to demonstrate the quality of refinement and deserves more attention in my view.

REVIEWER COMMENTS

Reviewer #1 (Remarks to the Author):

The authors developed enhanced cryo-cRED by combining hierarchical cluster analysis with cryo-EM for atomic level structural analysis of several 3D COFs. High quality 3D ED data were collected to obtain the ab initio structure solutions of five complex 3D COFs. All of non-hydrogen atoms were successfully located. This work is well presented and will benefit the development of COFs. I recommend the publication of this work after addressing the following concerns.

Thanks very much for the comments. We have tried our best to address reviewer's concerns and the following is our point-by-point response.

1. How was the cryo sample prepared? On page 5, "the sample was cooled to ~173K using a...". How was it achieved and what is the purpose? More detailed information should be provided.
Response: Thanks for the comment.

(1) The Cryo sample preparation has been described in our manuscript. Please refer to the first paragraph of the "3D flexible COFs with dynamic structures" and "TEM sample preparation" section. In short, the Cryo sample preparation is almost the same as the preparation of ordinary TEM samples except for loading the sample onto a Cryo transfer holder (Figure R1; Science 2017, 358, 506-510). The sample is thus kept in the low-temperature environment both outside and inside the transmission electron microscope (TEM) column by liquid nitrogen and the data collection only starts when the temperature of the sample is maintained at 96-99 K (the lowest sample temperature that our Cryo-transfer holder can reach inside the TEM).

(2) 173K is the lowest temperature that our cryo-transfer holder can obtain before inserting into the transmission electron microscope (TEM). The temperature of the sample can be detected by the temperature detector connected to the cryo-transfer holder during sample preparation and cRED data collection. Because of the high vacuum in TEM, the guest molecules in COFs are easy to escape. While FCOF-5 has a flexible structure, in order to obtain the expanded structure, guest molecules need to be fixed in the pores before the sample being inserted into the TEM.

Figure R1. Schematic Cryo sample preparation using a Cryo-transfer holder (Science 2017, 358, 506-510).

The TEM sample preparation section is revised as follows.

TEM Sample preparation

For E-FCOF-5, 3D-TPB-COF-OMe, 3D-TPB-COF-Me and 3D-TPB-COF-OH TEM samples were prepared by firstly crushing the powder followed by dispersion in ethanol and ultrasonic

treatment for five min. A drop of the dispersion was then deposited onto a 3 mm wide copper TEM grid covered by a carbon film. A high-tilt cryogenic sample holder equipped with temperature monitor (Gatan Company, Model 914) was used to freeze the sample to ~173 K to fix the guest molecular in pores before transferring the grid into TEM. The samples were further cooled to 96 K after inserting the holder in TEM during data collection. For C-FCOF-5, the tiny crystals were loaded directly onto the copper grid without dispersion in order to avoid adsorption of any guest molecular. To ensure that the framework is completely contracted, the sample is transferred into the TEM with high vacuum ($<2 \times 10^{-5}$ Pa) at room temperature for 5 min to further remove the guest molecules, which was then cooled to 96 K for data collection. All the data collection starts when the temperature of the sample was stable at 96-99K

2. There is a big difference among unit cell parameters from different datasets and Rint. What's the reason? How to deal with this during the merging of multiple datasets?

Response: The reason why there is a big difference among unit cell parameters from different datasets is because the distortion of electromagnetic lens in TEM is different when collecting data with different crystals. The difference of the Rint among different datasets resulted from the difference of crystal size and orientation. Multiple diffractions occur when electrons pass through the crystal, and different thicknesses and orientation of crystals have different degree of dynamic effects, resulting in different Rints.

For the merging of multiple datasets, the hierarchical cluster analysis (HCA) is performed to find the most common unit cell using the lattice-based clustering method and to select the optimal datasets for merge using the reflection-based clustering method (This is the key study to find which datasets should be merged for structure solution and refinement). XSCALE, one part of the XDS package, was used for merging the datasets and generating the hkl files that were used for structure solution and refinement. We used the averaged unit cell parameters of the selected datasets during the final refinement.

3. In the title, polycrystalline has been used. There is no need to use crystals afterwards. I think "Atomic-resolution structures from polycrystalline covalent organic frameworks with enhanced Cryo-cRED" is enough.

Response: Thanks for comments, the title has been revised as "Atomic-resolution structures from polycrystalline covalent organic frameworks with enhanced Cryo-cRED"

4. "ab initial" should be "ab initio".

Response: It has been revised as "ab initio"

5. In the abstract, anisotropic refinement is not a proper word. Refinement of anisotropic temperature factors or ADPs?

Response: It has been revised as "anisotropic temperature factors"

6. "The pursuit of...is to understand" may be changed to "The pursuit of...is a key to understand..."

Response: It has been revised as "is a key to understand relationship between structure and properties"

7. "With such powerful method" change to "With such a powerful method".

Response: It has been revised as "With such a powerful method"

8. Page 3, "(iv) uncertainties configuration" should be "uncertain configuration or configuration uncertainties".
Response: It has been revised as "configuration uncertainties"
9. "Due to the irreversible strong covalent bond properties," may be "due to the poor reversibility of strong covalent bonds"? Please confirm this.
Response: Yes, the authors agree with referee. It has been revised as "due to the poor reversibility of strong covalent bonds"
10. Page 4, "gust molecular" should be "guest molecule"?
Response: Thanks very much for the carefully reading. It has been revised as "guest molecule"
11. On page 5, "To achieve high diffraction resolution, the spot size and exposure time were optimized to 3 and 0.5s." The electron dose of spot size 3 is not low, why not using spot size 5 considering the reduce of beam damage? What is the rough electron dose or electron dose rate under this condition?
*Response: Thanks very much for the comments.
Yes, the spot size 3 is not the smallest one for the Jeol 2100 TEM. The spot size and exposure time that we used are the result of our multiple experimental optimizations. The beam damage would be future reduced using spot size but will result in a lower diffraction resolution, as the signal to noise ratio will decrease with the lower electron dose at high 2theta angle. So, we used spot size 3 to balance the resolution and beam damage. Regarding the electron dose under our experiment condition, we do not have the accurate value of electron dose as we do not use direct electron cameras with electron-counting function. In general, the dose rate was kept at lower than $0.1 \text{ e s}^{-1} \text{ \AA}^{-2}$ during data collection (please kindly refer to *Coordin. Chem. Rev.* 2021, 427, 213583).*
12. Unit cell parameter symbols should be italic, such as a, b, c, and beta.
Response: It has been revised accordingly.
13. "direct method" should be "direct methods".
Response: It has been revised as "direct methods"
14. On page 6, "The twenty-two cRED datasets could be...a = 13,7095(2)...", unit cell parameters have four decimal points and uncertainties were give. How can these parameters be so precise?
*Response: Thanks very much for the comments.
After merging the selected 16 datasets with high resolution ($<1\text{\AA}$) and high completeness ($>90\%$), a large number of data points are used to determine the unit cell parameters, which would be resulted in precise parameters.*
15. Could the number of ethanol molecules be precisely confirmed? Are there any unique interaction between ethanol and the framework?
Response: Thanks for the comments. The number of ethanol molecules can not be precisely confirmed as the residual factors (R factor) with refinement against on cRED data are a little bit high compared with single crystal X-ray diffraction data, which resulted from the dynamic effects of electron diffraction. We could not get a precise value of occupancy for the guest molecules. But the position of the guest molecules can be easily obtained from the different electrostatic potential map during the refinement. Our study shown that the ethanol or water in the pores have very weak hydrogen bond interaction with the framework.

16. On page 8, "To ensure that..." Could this process completely remove the guest molecules?
Response, Yes, this process (a high vacuum treatment within the TEM for five minutes) can remove the guest molecules completely, as there is no residual electron density in the closed pores for C-FCOF-5.
17. On page 11, "the atomic precession structure" should be "the atomic precision structure".
Response: It has been revised as "the atomic precision structure"
18. As the reported standard Cryo-EM sample preparation, standard fast-cooling method can keep the room-temperature structure (J. Am. Chem. Soc. 2019, 141, 19983–19987; doi:10.1038/s41467-022-29332-2). Can this cooling process keep the innate structure? As a flexible COF, dose cooling temperature influence the structure, especially for C-FCOF-5?
*Response: The references mentioned by the reviewers mainly focus on the preparation of protein crystals. The protein structure has a multi-level structure and is very sensitive to changes in the external environment, especially the tertiary and the quaternary structure. The stability of the protein tertiary and quaternary structure mainly depends on secondary bonds, including hydrogen bonds, hydrophobic bonds, salt bonds and van der Waals forces with weak interaction. In order to keep the room-temperature structure (innate structure), the standard fast-cooling method for protein sample is using liquid ethane which is faster than liquid nitrogen. Water or water vapor at room temperature is amorphous, and rapid freezing can maintain the amorphous state on the surface of protein crystals, which will benefit for subsequent ED data collection and reduction.
Of course, the standard fast-cooling method for protein materials also can be applied on inorganic small molecule compounds. For the FCOF-5, the framework is made up of very stable covalent bonds, the completely expansion and contraction state can be easily obtained and very stable under room temperature before cooling, so the expansion and contraction structure of FCOF-5 will not change during cooling.*
19. On page 14, some spelling errors, such as "need to be process", "is run", "for merge" et al. Please correct these.
Response: Thanks very much for the carefully review and we are sorry for these errors. As suggested, we have corrected all of these errors in the revised manuscript as "need to be processed", " will run", and "for merging", along with other corrections.
20. Please check the manuscript carefully and correct the typos.
Response: We sincerely thank the reviewer for his/her appraisal of our work and his/her comments, and are sorry for the typos in our manuscript. We have carefully checked our manuscript and corrected the typos in our manuscript. Please refer to our revised manuscript for details, where the revision is highlighted in yellow background.

Reviewer #2 (Remarks to the Author):

This is an important contribution towards the structural determination of COF materials. This is among the most challenging structural analyses. Porous COF materials have attracted much interests since

the discovery for a wide range of applications. However, due to their poorly crystalline nature, highly dynamic bonds/framework flexibility, unambiguous structural characterisation of these materials has been extremely difficult, if not possible. The growth of single crystals of COFs that are suitable for routine single crystal diffraction has been only a couple of examples, while analysis by powder diffraction is limited by its intrinsic low resolution and heavy overlap of peaks, which is particularly the case for COFs given their complex structures. The present study has developed Cryo-cRED to determine the crystal structures of a series of complex COFs. I am impressed by the quality of the datasets and refined structures; these are almost comparable with routine single crystals with resolutions down to 0.8 Å and completeness above 90%. This is a major achievement considering the polycrystalline nature and sensitivity to beam damage of these materials!

The structural analysis has been described in great details and new structural information revealed for some of these previously known COFs. I am particularly pleased to see that the team were able to observe isolated guest molecules in the pore by this technique. This I think is a big step forward in the structural characterisation of organic polymers. I therefore strongly recommend for publication of this work in Nature Commun.

Response, We thank the reviewer for his/her appraisal of our work and his/her comments.

One minor point, the authors could consider to move the tables of refinement details from the SI to the main paper. This contains some important structural information to demonstrate the quality of refinement and deserves more attention in my view.

Response, Thanks for the suggestion. Because the space in the main text is limited, the authors think it is better to keep the tables in SI.

Reviewers' Comments:

Reviewer #1:

Remarks to the Author:

My comments and concerns have been addressed in the revised manuscript. It is fine for me to publish as it is.

Response to Reviewer's Comments

REVIEWERS' COMMENTS

Reviewer #1 (Remarks to the Author):

My comments and concerns have been addressed in the revised manuscript. It is fine for me to publish as it is.

Response: We appreciate the reviewer for the comments and suggestions, as well as the efforts in improving the quality of our manuscript.